# Exploiting the Intrinsic Neighborhood Structure for Source-free Domain Adaptation

**Shiqi Yang**[1], **Yaxing Wang**[1,2]*, **Joost van de Weijer**[1], **Luis Herranz**[1], **Shangling Jui**[3]

[1] Computer Vision Center, Universitat Autonoma de Barcelona, Barcelona, Spain
[2] PCALab, Nanjing University of Science and Technology, China
[3] Huawei Kirin Solution, Shanghai, China

{syang,yaxing,joost,lherranz}@cvc.uab.es, jui.shangling@huawei.com

## Abstract

Domain adaptation (DA) aims to alleviate the domain shift between source domain and target domain. Most DA methods require access to the source data, but often that is not possible (e.g. due to data privacy or intellectual property). In this paper, we address the challenging source-free domain adaptation (SFDA) problem, where the source pretrained model is adapted to the target domain in the absence of source data. Our method is based on the observation that target data, which might no longer align with the source domain classifier, still forms clear clusters. We capture this intrinsic structure by defining local affinity of the target data, and encourage label consistency among data with high local affinity. We observe that higher affinity should be assigned to reciprocal neighbors, and propose a self regularization loss to decrease the negative impact of noisy neighbors. Furthermore, to aggregate information with more context, we consider expanded neighborhoods with small affinity values. In the experimental results we verify that the inherent structure of the target features is an important source of information for domain adaptation. We demonstrate that this local structure can be efficiently captured by considering the local neighbors, the reciprocal neighbors, and the expanded neighborhood. Finally, we achieve state-of-the-art performance on several 2D image and 3D point cloud recognition datasets. Code is available in https://github.com/Albert0147/SFDA_neighbors.

## 1 Introduction

Most deep learning methods rely on training on large amount of labeled data, while they cannot generalize well to a related yet different domain. One research direction to address this issue is Domain Adaptation (DA), which aims to transfer learned knowledge from a source to a target domain. Most existing DA methods demand labeled source data during the adaptation period, however, it is often not practical that source data are always accessible, such as when applied on data with privacy or property restrictions. Therefore, recently, there have emerged a few works [16, 17, 20, 21] tackling a new challenging DA scenario where instead of source data only the source pretrained model is available for adapting, *i.e.*, source-free domain adaptation (SFDA). Among these methods, USFDA [16] addresses universal DA [57] and SF [17] addresses open-set DA [36]. In both universal and open-set DA the label set is different for source and target domains. SHOT [21] and 3C-GAN [20] are for closed-set DA where source and target domains have the same categories. 3C-GAN [20] is based on target-style image generation with a conditional GAN, and SHOT [21] is based on mutual information maximization and pseudo labeling. Finally, BAIT [56] extends MCD [35] to the SFDA

---

*Corresponding Author.

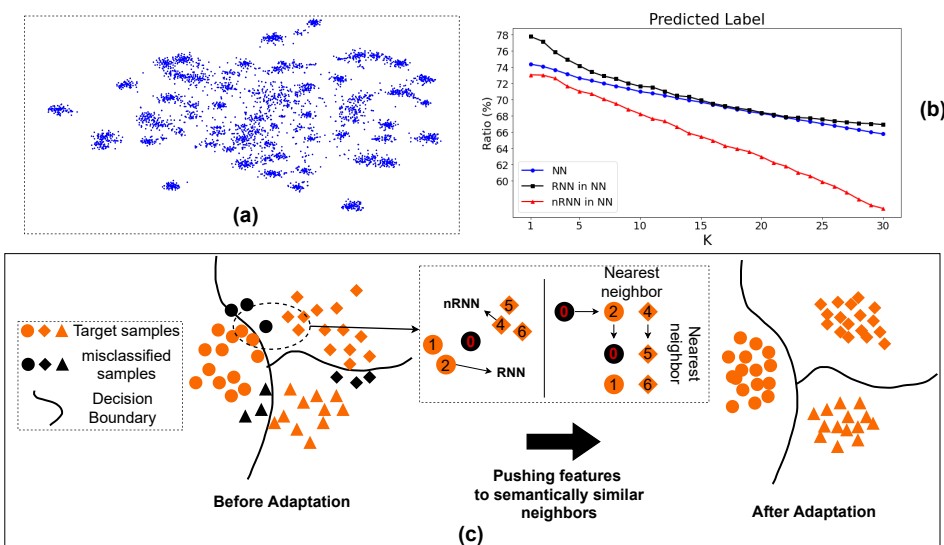

Figure 1: (**a**) t-SNE visualization of target features by source model. (**b**) Ratio of different type of nearest neighbor features of which: the *predicted* label is the same as the feature, K is the number of nearest neighbors. The features in (a) and (b) are on task Ar→Rw of Office-Home. (**c**) Illustration of our method. In the left shows we distinguish reciprocal and non-reciprocal neighbors. The adaptation is achieved by pushed the features towards reciprocal neighbors heavily.

setting. However, these methods ignore the intrinsic neighborhood structure of the target data in feature space which can be very valuable to tackle SFDA.

In this paper, we focus on closed-set source-free domain adaptation. Our main observation is that current DA methods do not exploit the intrinsic neighborhood structure of the target data. We use this term to refer to the fact that, even though the target data might have shifted in the feature space (due to the covariance shift), target data of the same class is still expected to form a cluster in the embedding space. This can be implied to some degree from the t-SNE visualization of target features on the source model which suggests that significant cluster structure is preserved (see Fig. 1 (a)). This assumption is implicitly adopted by most DA methods, as instantiated by a recent DA work [42]. A well-established way to assess the structure of points in high-dimensional spaces is by considering the nearest neighbors of points, which are expected to belong to the same class. However, this assumption is not true for all points; the blue curve in Figure 1(b) shows that around 75% of the nearest neighbors has the correct label. In this paper, we observe that this problem can be mitigated by considering reciprocal nearest neighbors (RNN); the reciprocal neighbors of a point have the point as their neighbor. Reciprocal neighbors have been studied before in different contexts [14, 31, 60]. The reason why reciprocal neighbors are more trustworthy is illustrated in Fig. 1(c). Fig. 1(b) shows the ratio of neighbors which have the *correct prediction* for different kinds of nearest neighbors. The curves show that reciprocal neighbors indeed have more chances to predict the *true* label than non-reciprocal nearest neighbors (nRNN).

The above observation and analysis motivate us to assign different weights to the supervision from nearest neighbors. Our method, called Neighborhood Reciprocity Clustering (*NRC*), achieves source-free domain adaptation by encouraging reciprocal neighbors to concord in their label prediction. In addition, we will also consider a weaker connection to the non-reciprocal neighbors. We define affinity values to describe the degree of connectivity between each data point and its neighbors, which is also utilized to encourage class-consistency between neighbors, and we propose to use a self-regularization to decrease the negative impact of potential noisy neighbors. Furthermore, inspired by recent graph based methods [1, 3, 61] which show that the higher order neighbors can provide relevant context, and also considering neighbors of neighbors is more likely to provide datapoints that are close on the data manifold [43]. Thus, to aggregate wider local information, we further retrieve the expanded neighbors, *i.e*, neighbor of the nearest neighbors, for auxiliary supervision.

Our contributions can be summarized as follows, to achieve source-free domain adaptation: (i) we explicitly exploit the fact that same-class data forms cluster in the target embedding space, we do

this by considering the predictions of neighbors and reciprocal neighbors, (ii) we further show that considering an extended neighborhood of data points further improves results (iii) the experiments results on three 2D image datasets and one 3D point cloud dataset show that our method achieves state-of-the-art performance compared with related methods.

## 2    Related Work

**Domain Adaptation.** Most DA methods tackle domain shift by aligning the feature distributions. Early DA methods such as [23, 41, 45] adopt moment matching to align feature distributions. And in recent years, plenty of works have emerged that achieve alignment by adversarial training. DANN [7] formulates domain adaptation as an adversarial two-player game. The adversarial training of CDAN [24] is conditioned on several sources of information. DIRT-T [40] performs domain adversarial training with an added term that penalizes violations of the cluster assumption. Additionally, [18, 26, 35] adopts prediction diversity between multiple learnable classifiers to achieve local or category-level feature alignment between source and target domains. AFN [52] shows that the erratic discrimination of target features stems from much smaller norms than those found in the source features. SRDC [42] proposes to directly uncover the intrinsic target discrimination via discriminative clustering to achieve adaptation. More related, [27] resorts to K-means clustering for open-set adaptation while considering global structure. Our method instead only focuses on nearest neighbors (local structure) for source-free adaptation.

**Source-free Domain Adaptation.** Source-present methods need supervision from the source domain during adaptation. Recently, there are several methods investigating source-free domain adaptation. USFDA [16] and FS [17] explore source-free universal DA [57] and open-set DA [36], and they propose to synthesize extra training samples to make the decision boundary compact, thereby allowing to recognise the open classes. For closed-set DA setting. SHOT [21] proposes to fix the source classifier and match the target features to the fixed classifier by maximizing mutual information and a proposed pseudo label strategy which considers global structure. 3C-GAN [20] synthesizes labeled target-style training images based on the conditional GAN to provide supervision for adaptation. Finally, SFDA [22] is for segmentation based on synthesizing fake source samples.

**Graph Clustering.** Our method shares some similarities with graph clustering work such as [38, 48, 54, 55] by utilizing neighborhood information. However, our methods are fundamentally different. Unlike those works which require labeled data to train the graph network for estimating the affinity, we instead adopt reciprocity to assign affinity.

## 3    Method

**Notation.** We denote the labeled source domain data with $n_s$ samples as $\mathcal{D}_s = \{(x_i^s, y_i^s)\}_{i=1}^{n_s}$, where the $y_i^s$ is the corresponding label of $x_i^s$, and the unlabeled target domain data with $n_t$ samples as $\mathcal{D}_t = \{x_j^t\}_{j=1}^{n_t}$. Both domains have the same $C$ classes (closed-set setting). Under the SFDA setting $\mathcal{D}_s$ is only available for model pretraining. Our method is based on a neural network, which we split into two parts: a feature extractor $f$, and a classifier $g$. The feature output by the feature extractor is denoted as $z(x) = f(x)$, the output of network is denoted as $p(x) = \delta(g(z)) \in \mathcal{R}^C$ where $\delta$ is the softmax function, for readability we will abandon the input and use $z, p$ in the following sections.

**Overview.** We assume that the source pretrained model has already been trained. As discusses in the introduction, the target features output by the source model form clusters. We exploit this intrinsic structure of the target data for SFDA by considering the neighborhood information, and the adaptation is achieved with the following objective:

$$\mathcal{L} = -\frac{1}{n_t} \sum_{x_i \in \mathcal{D}_t} \sum_{x_j \in \text{Neigh}(x_i)} \frac{D_{sim}(p_i, p_j)}{D_{dis}(x_i, x_j)} \tag{1}$$

where the $\text{Neigh}(x_i)$ means the nearest neighbors of $x_i$, $D_{sim}$ computes the similarity between predictions, and $D_{dis}$ is a constant measuring the semantic distance (dissimilarity) between data. The principle behind the objective is to push the data towards their semantically close neighbors by encouraging similar predictions. In the next sections, we will define $D_{sim}$ and $D_{dis}$.

## 3.1 Encouraging Class-Consistency with Neighborhood Affinity

To achieve adaptation without source data, we use the prediction of the nearest neighbor to encourage prediction consistency. While the target features from the source model are not necessarily totally intrinsic discriminative, meaning some neighbors belong to different class and will provide the wrong supervision. To decrease the potentially negative impact of those neighbors, we propose to weigh the supervision from neighbors according to the connectivity (semantic similarity). We define *affinity* values to signify the connectivity between the neighbor and the feature, which corresponds to the $\frac{1}{D_{dis}}$ in Eq. 1 indicating the semantic similarity.

To retrieve the nearest neighbors for batch training, similar to [33, 50, 62], we build two memory banks: $\mathcal{F}$ stores all target features, and $\mathcal{S}$ stores corresponding prediction scores:

$$\mathcal{F} = [\boldsymbol{z}_1, \boldsymbol{z}_2, \dots, \boldsymbol{z}_{n_t}] \text{and } \mathcal{S} = [p_1, p_2, \dots, p_{n_t}] \tag{2}$$

We use the cosine similarity for nearest neighbors retrieving. The difference between ours and [33, 50] lies in the fact that we utilize the memory bank to retrieve nearest neighbors while [33, 50] adopts the memory bank to compute the instance discrimination loss. Before every mini-batch training, we simply update the old items in the memory banks corresponding to current mini-batch. Note that updating the memory bank is only done to replace the old low-dimension vectors with new ones computed by the model, and does not require any additional computation.

We then use the prediction of the neighbors to supervise the training weighted by the affinity values, with the following objective adapted from Eq. 1:

$$\mathcal{L}_{\mathcal{N}} = -\frac{1}{n_t} \sum_i \sum_{k \in \mathcal{N}_K^i} A_{ik} \mathcal{S}_k^\top p_i \tag{3}$$

where we use the dot product to compute the similarity between predictions, corresponding to $D_{sim}$ in Eq.1, the $k$ is the index of the $k$-th nearest neighbors of $\boldsymbol{z}_i$, $\mathcal{S}_k$ is the $k$-th item in memory bank $\mathcal{S}$, $A_{ik}$ is the affinity value of $k$-th nearest neighbors of feature $\boldsymbol{z}_i$. Here the $\mathcal{N}_K^i$ is the index set[2] of the $K$-nearest neighbors of feature $\boldsymbol{z}_i$. Note that all neighbors are retrieved from the feature bank $\mathcal{F}$. With the affinity value as weight, this objective pushes the features to their neighbors with strong connectivity and to a lesser degree to those with weak connectivity.

To assign larger affinity values to semantic similar neighbors, we divide the nearest neighbors retrieved into two groups: reciprocal nearest neighbors (RNN) and non-reciprocal nearest neighbors (nRNN). The feature $\boldsymbol{z}_j$ is regarded as the RNN of the feature $\boldsymbol{z}_i$ if it meets the following condition:

$$j \in \mathcal{N}_K^i \land i \in \mathcal{N}_M^j \tag{4}$$

Other neighbors which do not meet the above condition are nRNN. Note that the normal definition of reciprocal nearest neighbors [31] applies $K = M$, while in this paper $K$ and $M$ can be different. We find that reciprocal neighbors have a higher potential to belong to the same cluster as the feature (Fig. 1(b)). Thus, we assign a high affinity value to the RNN features. Specifically for feature $\boldsymbol{z}_i$, the affinity value of its $j$-th K-nearest neighbor is defined as:

$$A_{i,j} = \begin{cases} 1 & \text{if } j \in \mathcal{N}_K^i \land i \in \mathcal{N}_M^j \\ r & \text{otherwise.} \end{cases} \tag{5}$$

where $r$ is a hyperparameter. If not specified $r$ is set to 0.1.

To further reduce the potential impact of noisy neighbors in $\mathcal{N}_K$, which belong to the different class but still are RNN, we propose a simply yet effective way dubbed *self-regularization*, that is, to not ignore the current prediction of ego feature:

$$\mathcal{L}_{self} = -\frac{1}{n_t} \sum_i^{n_t} \mathcal{S}_i^\top p_i \tag{6}$$

where $\mathcal{S}_i$ means the stored prediction in the memory bank, note this term is a *constant vector* and is identical to the $p_i$ since we update the memory banks before the training, here the loss is only back-propagated for variable $p_i$.

---

[2]All indexes are in the same order for the dataset and memory banks.

**Algorithm 1** Neighborhood Reciprocity Clustering for Source-free Domain Adaptation

---

**Require:** $\mathcal{D}_s$ (only for source model training), $\mathcal{D}_t$
 1: Pre-train model on $\mathcal{D}_s$
 2: Build feature bank $\mathcal{F}$ and score bank $\mathcal{S}$ for $\mathcal{D}_t$
 3: **while** Adaptation **do**
 4:     Sample batch $\mathcal{T}$ from $\mathcal{D}_t$
 5:     Update $\mathcal{F}$ and $\mathcal{S}$ corresponding to current batch $\mathcal{T}$
 6:     Retrieve nearest neighbors $\mathcal{N}$ for each of $\mathcal{T}$
 7:     Compute affinity value $A$                                        ▷ Eq.5
 8:     Retrieve expanded neighborhoods $E$ for each of $\mathcal{N}$
 9:     Compute loss and update the model                                ▷ Eq. 9
10: **end while**

---

To avoid the degenerated solution [8, 39] where the model predicts all data as some specific classes (and does not predict other classes for any of the target data), we encourage the prediction to be balanced. We adopt the prediction diversity loss which is widely used in clustering [8, 9, 13] and also in several domain adaptation works [21, 39, 42]:

$$\mathcal{L}_{div} = \sum_{c=1}^{C} \mathrm{KL}(\bar{p}_c \| q_c), \text{with } \bar{p}_c = \frac{1}{n_t} \sum_i p_i^{(c)}, \text{and } q_{\{c=1,..,C\}} = \frac{1}{C} \tag{7}$$

where the $p_i^{(c)}$ is the score of the $c$-th class and $\bar{p}_c$ is the empirical label distribution, it represents the predicted possibility of class $c$ and q is a uniform distribution.

### 3.2  Expanded Neighborhood Affinity

As mentioned in Sec. 1, a simple way to achieve the aggregation of more information is by considering more nearest neighbors. However, a drawback is that larger neighborhoods are expected to contain more datapoint from multiple classes, defying the purpose of class consistency. A better way to include more target features is by considering the $M$-nearest neighbor of each neighbor in $\mathcal{N}_K$ of $z_i$ in Eq. 4, *i.e.*, the expanded neighbors. These target features are expected to be closer on the target data manifold than the features that are included by considering a larger number of nearest neighbors [43]. The expanded neighbors of feature $z_i$ are defined as $E_M(z_i) = \mathcal{N}_M(z_j) \, \forall j \in \mathcal{N}_K(z_i)$, *note that $E_M(z_i)$ is still an index set and $i$ (ego feature) $\notin E_M(z_i)$*. We directly assign a small affinity value $r$ to those expanded neighbors, since they are further than nearest neighbors and may contain noise. We utilize the prediction of those expanded neighborhoods for training:

$$\mathcal{L}_E = -\frac{1}{n_t} \sum_i \sum_{k \in \mathcal{N}_K^i} \sum_{m \in E_M^k} r \mathcal{S}_m^\top p_i \tag{8}$$

where $E_M^k$ contain the $M$-nearest neighbors of neighbor $k$ in $\mathcal{N}_K$.

Although the affinity values of all expanded neighbors are the same, it does not necessarily mean that they have equal importance. Taking a closer look at the expanded neighbors $E_M(z_i)$, some neighbors will show up more than once, for example $z_m$ can be the nearest neighbor of both $z_h$ and $z_j$ where $h, j \in \mathcal{N}_K(z_i)$, and the nearest neighbors can also serve as expanded neighbor. It implies that those neighbors form compact cluster, and we posit that those duplicated expanded neighbors have potential to be semantically closer to the ego-feature $z_i$. Thus, we do not remove duplicated features in $E_M(z_i)$, as those can lead to actually larger affinity value for those expanded neighbors. This is one advantage of utilizing expanded neighbors instead of more nearest neighbors, we will verify the importance of maintaining the duplicated features in the experimental section.

**Final objective.** Our method, called *Neighborhood Reciprocity Clustering* (*NRC*), is illustrated in Algorithm. 1. The final objective for adaptation is:

$$\mathcal{L} = \mathcal{L}_{div} + \mathcal{L}_{\mathcal{N}} + \mathcal{L}_E + \mathcal{L}_{self}. \tag{9}$$

## 4  Experiments

**Datasets.** We use three 2D image benchmark datasets and a 3D point cloud recognition dataset. **Office-31** [32] contains 3 domains (Amazon, Webcam, DSLR) with 31 classes and 4,652 images.

Table 1: Accuracies (%) on Office-31 for ResNet50-based methods.

| Method | SF | A→D | A→W | D→W | W→D | D→A | W→A | Avg |
|---|---|---|---|---|---|---|---|---|
| MCD [35] | ✗ | 92.2 | 88.6 | 98.5 | **100.0** | 69.5 | 69.7 | 86.5 |
| CDAN [24] | ✗ | 92.9 | 94.1 | 98.6 | **100.0** | 71.0 | 69.3 | 87.7 |
| MDD [59] | ✗ | 90.4 | 90.4 | 98.7 | 99.9 | 75.0 | 73.7 | 88.0 |
| BNM [4] | ✗ | 90.3 | 91.5 | 98.5 | **100.0** | 70.9 | 71.6 | 87.1 |
| DMRL [49] | ✗ | 93.4 | 90.8 | 99.0 | **100.0** | 73.0 | 71.2 | 87.9 |
| BDG [53] | ✗ | 93.6 | 93.6 | 99.0 | **100.0** | 73.2 | 72.0 | 88.5 |
| MCC [15] | ✗ | 95.6 | 95.4 | 98.6 | 100.0 | 72.6 | 73.9 | 89.4 |
| SRDC [42] | ✗ | 95.8 | 95.7 | 99.2 | 100.0 | 76.7 | 77.1 | 90.8 |
| RWOT [51] | ✗ | 94.5 | 95.1 | **99.5** | 100.0 | **77.5** | 77.9 | 90.8 |
| RSDA-MSTN [10] | ✗ | 95.8 | **96.1** | 99.3 | **100.0** | 77.4 | **78.9** | **91.1** |
| SHOT [21] | ✓ | 94.0 | 90.1 | 98.4 | 99.9 | 74.7 | 74.3 | 88.6 |
| 3C-GAN [20] | ✓ | 92.7 | 93.7 | 98.5 | 99.8 | 75.3 | 77.8 | 89.6 |
| **NRC** | ✓ | **96.0** | 90.8 | 99.0 | **100.0** | 75.3 | 75.0 | 89.4 |

Table 2: Accuracies (%) on Office-Home for ResNet50-based methods.

| Method | SF | Ar→Cl | Ar→Pr | Ar→Rw | Cl→Ar | Cl→Pr | Cl→Rw | Pr→Ar | Pr→Cl | Pr→Rw | Rw→Ar | Rw→Cl | Rw→Pr | **Avg** |
|---|---|---|---|---|---|---|---|---|---|---|---|---|---|---|
| MCD [35] | ✗ | 48.9 | 68.3 | 74.6 | 61.3 | 67.6 | 68.8 | 57.0 | 47.1 | 75.1 | 69.1 | 52.2 | 79.6 | 64.1 |
| CDAN [24] | ✗ | 50.7 | 70.6 | 76.0 | 57.6 | 70.0 | 70.0 | 57.4 | 50.9 | 77.3 | 70.9 | 56.7 | 81.6 | 65.8 |
| SAFN [52] | ✗ | 52.0 | 71.7 | 76.3 | 64.2 | 69.9 | 71.9 | 63.7 | 51.4 | 77.1 | 70.9 | 57.1 | 81.5 | 67.3 |
| Symnets [58] | ✗ | 47.7 | 72.9 | 78.5 | 64.2 | 71.3 | 74.2 | 64.2 | 48.8 | 79.5 | 74.5 | 52.6 | 82.7 | 67.6 |
| MDD [59] | ✗ | 54.9 | 73.7 | 77.8 | 60.0 | 71.4 | 71.8 | 61.2 | 53.6 | 78.1 | 72.5 | **60.2** | 82.3 | 68.1 |
| TADA [47] | ✗ | 53.1 | 72.3 | 77.2 | 59.1 | 71.2 | 72.1 | 59.7 | 53.1 | 78.4 | 72.4 | 60.0 | 82.9 | 67.6 |
| BNM [4] | ✗ | 52.3 | 73.9 | 80.0 | 63.3 | 72.9 | 74.9 | 61.7 | 49.5 | 79.7 | 70.5 | 53.6 | 82.2 | 67.9 |
| BDG [53] | ✗ | 51.5 | 73.4 | 78.7 | 65.3 | 71.5 | 73.7 | 65.1 | 49.7 | 81.1 | 74.6 | 55.1 | 84.8 | 68.7 |
| SRDC [42] | ✗ | 52.3 | 76.3 | 81.0 | **69.5** | 76.2 | 78.0 | **68.7** | 53.8 | 81.7 | **76.3** | 57.1 | 85.0 | 71.3 |
| RSDA-MSTN [10] | ✗ | 53.2 | 77.7 | 81.3 | 66.4 | 74.0 | 76.5 | 67.9 | 53.0 | 82.0 | 75.8 | 57.8 | 85.4 | 70.9 |
| SHOT [21] | ✓ | 57.1 | 78.1 | 81.5 | 68.0 | 78.2 | 78.1 | 67.4 | 54.9 | 82.2 | 73.3 | 58.8 | 84.3 | 71.8 |
| **NRC** | ✓ | **57.7** | **80.3** | **82.0** | 68.1 | **79.8** | **78.6** | 65.3 | **56.4** | **83.0** | 71.0 | 58.6 | **85.6** | **72.2** |

**Office-Home** [46] contains 4 domains (Real, Clipart, Art, Product) with 65 classes and a total of 15,500 images. **VisDA** [28] is a more challenging dataset, with 12-class synthetic-to-real object recognition tasks, its source domain contains of 152k synthetic images while the target domain has 55k real object images. **PointDA-10** [30] is the first 3D point cloud benchmark specifically designed for domain adaptation, it has 3 domains with 10 classes, denoted as ModelNet-10, ShapeNet-10 and ScanNet-10, containing approximately 27.7k training and 5.1k testing images together.

**Evaluation.** We compare with existing source-present and source-free DA methods. *All results are the average on three random runs.* **SF** in the tables denotes source-free.

**Model details.** For fair comparison with related methods, we also adopt the backbone of ResNet-50 [11] for Office-Home and ResNet-101 for VisDA, and PointNet [29] for PointDA-10. Specifically, for 2D image datasets, we use the same network architecture as SHOT [21], *i.e.*, the final part of the network is: fully connected layer − Batch Normalization [12] − fully connected layer with weight normalization [37]. And for PointDA-10 [29], we use the code released by the authors for fair comparison with PointDAN [29], and only use the backbone without any of their proposed modules. To train the source model, we also adopt label smoothing as SHOT does. We adopt SGD with momentum 0.9 and batch size of 64 for all 2D datasets, and Adam for PointDA-10. The learning rate for Office-31 and Office-Home is set to 1e-3 for all layers, except for the last two newly added fc layers, where we apply 1e-2. Learning rates are set 10 times smaller for VisDA. Learning rate for PointDA-10 is set to 1e-6. We train 30 epochs for Office-31 and Office-Home while 15 epochs for VisDA, and 100 for PointDA-10. For the number of nearest neighbors (K) and expanded neighborhoods (M), we use 3,2 for Office-31, Office-Home and PointDA-10, since VisDA is much larger we set K, M to 5. Experiments are conducted on a TITAN Xp.

## 4.1 Results

**2D image datasets.** We first evaluate the target performance of our method compared with existing DA and SFDA methods on three 2D image datasets. As shown in Table 1-3, the top part shows results for the source-present methods *with access to source data during adaptation*. The bottom shows results for the source-free DA methods. On Office-31, our method gets similar results compared

Table 3: Accuracies (%) on VisDA-C (Synthesis → Real) for ResNet101-based methods.

| Method | SF | plane | bcycl | bus | car | horse | knife | mcycl | person | plant | sktbrd | train | truck | Per-class |
|---|---|---|---|---|---|---|---|---|---|---|---|---|---|---|
| ADR [34] | ✗ | 94.2 | 48.5 | 84.0 | 72.9 | 90.1 | 74.2 | 92.6 | 72.5 | 80.8 | 61.8 | 82.2 | 28.8 | 73.5 |
| CDAN [24] | ✗ | 85.2 | 66.9 | 83.0 | 50.8 | 84.2 | 74.9 | 88.1 | 74.5 | 83.4 | 76.0 | 81.9 | 38.0 | 73.9 |
| CDAN+BSP [2] | ✗ | 92.4 | 61.0 | 81.0 | 57.5 | 89.0 | 80.6 | 90.1 | 77.0 | 84.2 | 77.9 | 82.1 | 38.4 | 75.9 |
| SAFN [52] | ✗ | 93.6 | 61.3 | 84.1 | 70.6 | 94.1 | 79.0 | 91.8 | 79.6 | 89.9 | 55.6 | 89.0 | 24.4 | 76.1 |
| SWD [19] | ✗ | 90.8 | 82.5 | 81.7 | 70.5 | 91.7 | 69.5 | 86.3 | 77.5 | 87.4 | 63.6 | 85.6 | 29.2 | 76.4 |
| MDD [59] | ✗ | - | - | - | - | - | - | - | - | - | - | - | - | 74.6 |
| DMRL [49] | ✗ | - | - | - | - | - | - | - | - | - | - | - | - | 75.5 |
| MCC [15] | ✗ | 88.7 | 80.3 | 80.5 | 71.5 | 90.1 | 93.2 | 85.0 | 71.6 | 89.4 | 73.8 | 85.0 | 36.9 | 78.8 |
| STAR [26] | ✗ | 95.0 | 84.0 | **84.6** | 73.0 | 91.6 | 91.8 | 85.9 | 78.4 | 94.4 | 84.7 | 87.0 | 42.2 | 82.7 |
| RWOT [51] | ✗ | 95.1 | 80.3 | 83.7 | **90.0** | 92.4 | 68.0 | **92.5** | 82.2 | 87.9 | 78.4 | **90.4** | **68.2** | 84.0 |
| 3C-GAN [20] | ✓ | 94.8 | 73.4 | 68.8 | 74.8 | 93.1 | 95.4 | 88.6 | **84.7** | 89.1 | 84.7 | 83.5 | 48.1 | 81.6 |
| SHOT [21] | ✓ | 94.3 | 88.5 | 80.1 | 57.3 | 93.1 | 94.9 | 80.7 | 80.3 | 91.5 | 89.1 | 86.3 | 58.2 | 82.9 |
| **NRC** | ✓ | **96.8** | **91.3** | 82.4 | 62.4 | **96.2** | **95.9** | 86.1 | 80.6 | **94.8** | **94.1** | 90.4 | 59.7 | **85.9** |

Table 4: Accuracies (%) on PointDA-10. *The results except ours are from PointDA [30].*

| | SF | Model→Shape | Model→Scan | Shape→Model | Shape→Scan | Scan→Model | Scan→Shape | Avg |
|---|---|---|---|---|---|---|---|---|
| MMD [25] | ✗ | 57.5 | 27.9 | 40.7 | 26.7 | 47.3 | 54.8 | 42.5 |
| DANN [6] | ✗ | 58.7 | 29.4 | 42.3 | 30.5 | 48.1 | 56.7 | 44.2 |
| ADDA [44] | ✗ | 61.0 | 30.5 | 40.4 | 29.3 | 48.9 | 51.1 | 43.5 |
| MCD [35] | ✗ | 62.0 | 31.0 | 41.4 | 31.3 | 46.8 | 59.3 | 45.3 |
| PointDAN [30] | ✗ | 64.2 | **33.0** | 47.6 | **33.9** | 49.1 | 64.1 | 48.7 |
| Source-only | | 43.1 | 17.3 | 40.0 | 15.0 | 33.9 | 47.1 | 32.7 |
| **NRC** | ✓ | **64.8** | 25.8 | **59.8** | 26.9 | **70.1** | **68.1** | **52.6** |

with source-free method 3C-GAN and lower than source-present method RSDA-MSTN. And our method achieves state-of-the-art performance on Office-Home and VisDA, especially on VisDA our method surpasses the source-free method SHOT and source-present method RWOT by a wide margin (3% and 1.9% respectively). The reported results clearly demonstrate the efficiency of the proposed method for source-free domain adaptation. Interestingly, like already observed in the SHOT paper, source-free methods outperform methods that have access to source data during adaptation.

**3D point cloud dataset.** We also report the result for the PointDA-10. As shown in Table 4, our method outperforms PointDAN [30], which demands source data for adaptation and is specifically tailored for point cloud data with extra attention modules, by a large margin (4%).

## 4.2 Analysis

**Ablation study on neighbors $\mathcal{N}$, $E$ and affinity $A$.** In the first two tables of Table 5, we conduct the ablation study on Office-Home and VisDA. The 1-st row contains results from the source model and the 2-nd row from only training with the diversity loss $\mathcal{L}_{div}$. From the remaining rows, several conclusions can be drawn.

First, the original supervision, which considers all neighbors equally can lead to a decent performance (67.1 on Office-Home). Second, considering higher affinity values for reciprocal neighbors leads to a large performance gain (69.1 on Office-Home). Last but not the least, the expanded neighborhoods

Table 5: Ablation study of different modules on Office-Home (**left**) and VisDA (**middle**), comparison between using expanded neighbors and larger nearest neighbors (**right**).

| $\mathcal{L}_{div}$ | $\mathcal{L}_{\mathcal{N}}$ | $\mathcal{L}_E$ | $\mathcal{L}_{\hat{E}}$ | A | Avg |
|---|---|---|---|---|---|
| | | | | | 59.5 |
| ✓ | | | | | 62.1 |
| ✓ | ✓ | | | | 67.1 |
| ✓ | ✓ | | | ✓ | 69.1 |
| ✓ | ✓ | ✓ | | | 65.2 |
| ✓ | ✓ | ✓ | | ✓ | **72.2** |
| ✓ | ✓ | | ✓ | ✓ | 69.1 |

| $\mathcal{L}_{div}$ | $\mathcal{L}_{\mathcal{N}}$ | $\mathcal{L}_E$ | $\mathcal{L}_{\hat{E}}$ | A | Acc |
|---|---|---|---|---|---|
| | | | | | 44.6 |
| ✓ | | | | | 47.8 |
| ✓ | ✓ | | | | 74.6 |
| ✓ | ✓ | | | ✓ | 81.5 |
| ✓ | ✓ | ✓ | | | 61.2 |
| ✓ | ✓ | ✓ | | ✓ | **85.9** |
| ✓ | ✓ | | ✓ | ✓ | 82.0 |

| Method&Dataset | Acc |
|---|---|
| VisDA ($K=M=5$) | **85.9** |
| VisDA w/o $E$ ($K=30$) | 84.0 |
| OH ($K=3,M=2$) | **72.2** |
| OH w/o $E$ ($K=9$) | 69.5 |

Table 6: Runtime analysis on SHOT and our method. For SHOT, pseudo labels are computed at each epoch. 20%, 10% and 5% denote the percentage of target features which are stored in the memory bank.

| VisDA | Runtime (s/epoch) | Per-class (%) |
|---|---|---|
| SHOT | 618.82 | 82.9 |
| NRC | 540.89 | 85.9 |
| NRC(20% for memory bank) | 507.15 | 85.3 |
| NRC(10% for memory bank) | 499.49 | 85.2 |
| NRC(5% for memory bank) | 499.28 | 85.1 |

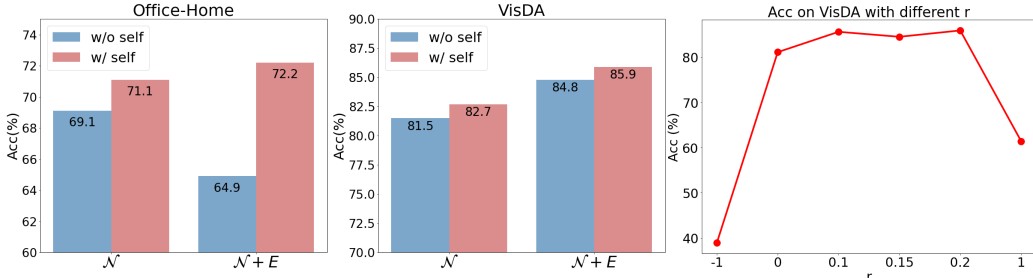

Figure 2: (**Left and middle**) Ablation study of $\mathcal{L}_{self}$ on Office-Home and VisDA respectively. (**Right**) Performance with different $r$ on VisDA.

can also be helpful, but only when combined with the affinity values $A$ (72.2 on Office-Home). Using expanded neighborhoods without affinity obtains bad performance (65,2 on Office-Home). We conjecture that those expanded neighborhoods, especially those neighbors of nRNN, may be noisy as discussed in Sec. 3.2. Removing the affinity $A$ means we treat all those neighbors equally, which is not reasonable.

We also show that duplication in the expanded neighbors is important in the last row of Table 5, where the $\mathcal{L}_{\hat{E}}$ means we remove duplication in Eq. 8. The results show that the performance will degrade significantly when removing them, implying that the duplicated expanded neighbors are indeed more important than others.

Next we ablate the importance of the expanded neighborhood in the right of Table5. We show that if we increase the number of datapoints considered for class-consistency by simply considering a larger K, we obtain significantly lower scores. We have chosen $K$ so that the total number of points considered is equal to our method (i.e. 5+5*5=30 and 3+3*2=9). Considering neighbors of neighbors is more likely to provide datapoints that are close on the data manifold [43], and are therefore more likely to share the class label with the ego feature.

**Runtime analysis.** Instead of storing all feature vectors in the memory bank, we follow the same memory bank setting as in [5] which is for nearest neighbor retrieval. The method only stores a fixed number of target features, we update the memory bank at the end of each iteration by taking the $n$ (batch size) embeddings from the current training iteration and concatenating them at the end of the memory bank, and discard the oldest $n$ elements from the memory bank. We report the results with this type of memory bank of different buffer size in the Table 6. The results show that indeed this could be an efficient way to reduce computation on very large datasets.

**Ablation study on self-regularization.** In the left and middle of Fig 2, we show the results with and without self-regularization $\mathcal{L}_{self}$. The $\mathcal{L}_{self}$ can improve the performance when adopting only nearest neighbors $\mathcal{N}$ or all neighbors $\mathcal{N} + E$. The results imply that self-regularization can effectively reduce the negative impact of the potential noisy neighbors, especially on the Office-Home dataset.

**Sensitivity to hyperparameter.** There are three hyperparameters in our method: K and M which are the number of nearest neighbors and expanded neighbors, $r$ which is the affinity value assigned to nRNN. We show the results with different $r$ in the right of Fig. 2. *Note we keep the affinity of expanded neighbors as 0.1*. $r = 1$ means no affinity. $r = -1$ means treating supervision of nRNN feature as totally wrong, which is not always the case and will lead to quite lower result. $r = 0$ can also achieve good performance, signifying RNN can already work well. Results with

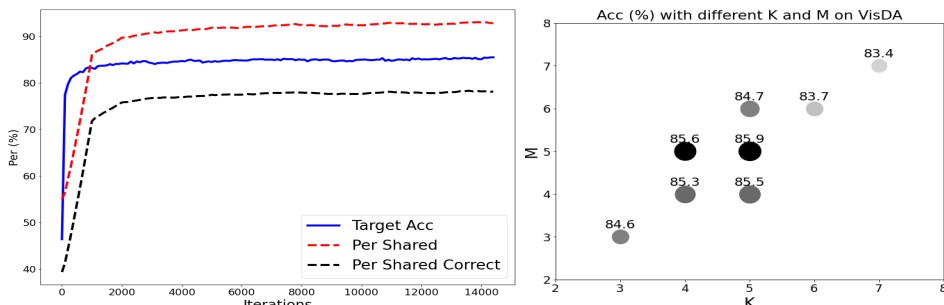

Figure 3: (**Left**) The three curves are (on VisDA): target accuracy (*Blue*), ratio of features which have 5-nearest neighbors all sharing the same predicted label (*dashed Red*), and ratio of features which have 5-nearest neighbors all sharing the same and *correct* predicted label (*dashed Black*). (**Right**) Ablation study on choice of K and M on VisDA.

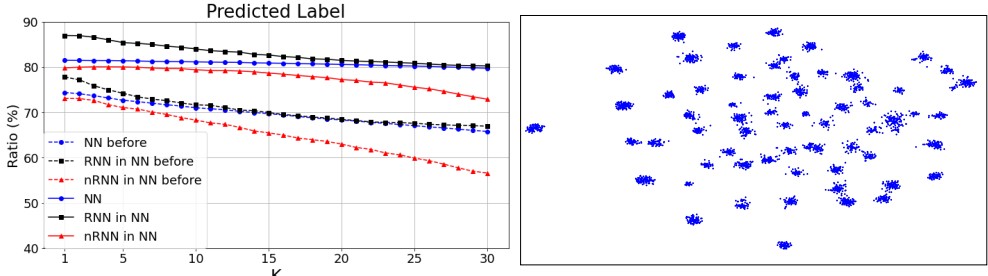

Figure 4: (**Left**) Ratio of different type of nearest neighbor features which have the correct predicted label, before and after adaptation. (**Right**) Visualization of target features after adaptation.

$r = 0.1/0.15/0.2$ show that our method is not sensitive to the choice of a reasonable $r$. Note in DA, there is no validation set for hyperparameter tuning, we show the results varying the number of neighbors in the right of Tab. 3, demonstrating the robustness to the choice of $K$ and $M$.

**Training curve.** We show the evolution of several statistics during adaptation on VisDA in the left of Tab. 3. The blue curve is the target accuracy. The dashed red and black curves are the ratio of features which have 5-nearest neighbors all sharing the same (*dashed Red*), or the same and also **correct** (*dashed Black*) predicted label. The curves show that the target features are clustering during the training. Another interesting finding is that the curve 'Per Shared' correlates with the accuracy curve, which might therefore be used to determine training convergence.

**Accuracy of supervision from neighbors.** We also show the accuracy of supervision from neighbors on task Ar→Rw of Office-Home in Fig. 4(left). It shows that after adaptation, the ratio of all types of neighbors having more correct predicted label, proving the effectiveness of the method.

**t-SNE visualization.** We show the t-SNE feature visualization on task Ar→Rw of target features before (Fig. 1(a)) and after (Fig. 4(right)) adaptation. After adaptation, the features are more compactly clustered.

## 5 Conclusions

We introduce a source-free domain adaptation (SFDA) method by uncovering the intrinsic target data structure. We propose to achieve the adaptation by encouraging label consistency among local target features. We differentiate between nearest neighbors, reciprocal neighbors and expanded neighborhood. Experimental results verify the importance of considering the local structure of the target features. Finally, our experimental results on both 2D image and 3D point cloud datasets testify the efficacy of our method.

**Acknowledgement** We acknowledge the support from Huawei Kirin Solution, and the project PID2019-104174GB-I00 (MINECO, Spain) and RTI2018-102285-A-I00 (MICINN, Spain), Ramón y Cajal fellowship RYC2019-027020-I, and the CERCA Programme of Generalitat de Catalunya.

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
