# OpenReview forum: "Exploiting the Intrinsic Neighborhood Structure for Source-free Domain Adaptation"
_NeurIPS.cc/2021/Conference — NeurIPS 2021 Poster_

### Official Review · Reviewer_tPBC · 2021-07-02

**Rating:** 7
**Confidence:** 4

**Summary:**

This work proposes a framework to perform Source-free Domain Adaptation by fully exploiting the local representation structure in the latent space. Specifically, it maintains two memory banks for logits and representations, respectively. Using the a sample in the mini-batch as query, it retrieve semantically similar neighborhoods from the bank, and constrains their logits to be similar. In addition, a self-regularization loss is applied to mitigate noisy neighbors, and a diversity loss is used to balance the sizes among different clusters. Experimental results on four benchmark datasets verify the superiority of the proposed method. Also, thorough ablation studies are conducted to analyze the effect of different model components.

**Limitations And Societal Impact:**

The limitations of the proposed method are less discussed in the current version. Also, the potential social impact is lacked in the paper. Authors are encouraged to analyze how their method can potentially benefit and also hamper the related real-world applications.

**Main Review:**

Strength:

1. The proposed framework is technically sound, which can exploit the local representation structure from various granularities.
2. The paper is well written, and it's easy to capture the gist of this work.
3. Sufficient experiments are conducted to evaluate the whole model and various model components.

Weakness and question:

1. Technically, I'm convinced that the proposed method is able to promote the feature compactness within some feature cluster. However, except for the feature compactness, it is also important to constrain the inter-cluster feature separability so as to guarantee the good performance on target domain. How can the proposed model achieves this important property?
2. In Eq. 1, authors use the distance towards query sample to depict semantics. Considering the representation bias of each specific instance, why not use the distance towards the cluster center associated to the query sample? This scheme may capture semantics more precisely.
3. The objectives in Eq. 9 are composed of a diversity term, two neighborhood semantic similarity terms and a self-regularization term. There should be a trade-off among these three kinds of objectives, such that some trade-off parameters might further promote the performance of this model.

Justification of rating:

I am convinced by the technical merit of this paper, and think it as a well-suited framework for Source-free Domain Adaptation. Though there are some points needed to be further clarified, this work is decent enough as a whole.


**Post Rebuttal**

I have read the response from authors and other reviews. In general, I admire authors' efforts on exploring the local and global neighborhoods structures in the representation space of multiple domains. Most of my concerns are addressed in the response, and thus I keep my positive rating.

**Time Spent Reviewing:**

2

---

> ### Author Response · Authors · 2021-08-10
> **Response to Reviewer tPBC**
>
> **1. How can the proposed model achieves this important property?**
>
> Thanks for pointing out that the inter-cluster separability is also important. In our main training objective, we do not have explicit effort to encourage inter cluster separability, since we do not actually consider any cluster specific information, such as class prototype. However we think the usage of reciprocity of neighbors may help to form clear cluster boundaries to some degree.  From the perspective of a graph (since our method is similar to a nearest neighbor graph), the connected nodes (in our case the feature and its neighbor) with different labels will impede the formation of clusters and make the inter-cluster boundary less clear [a]. While in our method, we give lower affinity to non-reciprocal neighbors which have higher possibility to belong to different classes than reciprocal ones, thus integrating less information of those potential noisy neighbors which may lead to less clear inter-cluster boundary.  Our method also includes the diversity loss. This loss encourages that all classes are predicted for some target data, which will result in inter-cluster feature separability.  The importance of this loss on the final performance (see Question 4 of Reviewer R8cb) reflects the importance of inter-cluster feature separability for DA.
>
> **2. Why not use the distance towards the cluster center associated to the query sample?**
>
> Indeed, the cluster prototype provides more global information than just using local neighbors. And SHOT actually considers this cluster prototype information to compute pseudo labels. In experiments our method surpasses SHOT on all datasets (3% higher than SHOT on VisDA). This may imply the noisy prediction/pseudo label may have a negative impact. We have also tried to directly combine this pseudo label with our method, thus including both local and global information, but it even gets lower results (1.5% lower on VisDA) compared to our method. Further research is required to investigate how to efficiently combine global and local information.
>
> **3. Some trade-off parameters in Eq. 9 might further promote the performance of this model.**
>
> Thanks for the advice, we will introduce these trade-off parameters in the final version. However, since we want to prevent complex parameter searches, we propose to set them to 1. We actually found that with a weight factor (0.5) applied to diversity loss, we can further improve results with 0.3%. Though, we expect a full parameter search to give some further gains, we prefer to present results with the basic setting where we use the same trade-off parameters on all datasets (setting them to 1).
>
> And we will add the Limitations And Societal Impact part in the paper.
>
> *reference:*
>
> [a] Hongwei Wang, and Jure Leskovec. "Unifying graph convolutional neural networks and label propagation." arXiv preprint arXiv:2002.06755 (2020).

---

### Official Review · Reviewer_4Suf · 2021-07-12

**Rating:** 4
**Confidence:** 4

**Summary:**

The authors propose a source-free domain adaptation (SFDA) method by exploiting the intrinsic target neighborhood structure. Extensive experiments conducted on both 2D image and 3D point cloud datasets demonstrate the effectiveness of the proposed method to a certain extent.

**Ethics Review Area:**

["I don’t know"]

**Main Review:**

1.	Please give more explanations and discussions about the motivation of utilizing the memory bank to retrieve nearest neighbors.
2.	Is the hyperparameter r used in Eq.5 and Eq.9 the same? Why not choose different hyperparameters for these two parts?
3.	Please provide comparative experiments with more source-free domain adaptation (SFDA) methods, such as [1] [2] [3].
4.	It’s recommended to provide visualization of target features before adaptation in Fig.5 for better comparison.
5.	Does the performance improve significantly at the expense of consuming large computational time cost? Please provide time complexity analysis of the proposed method.
6.    The current manuscript should be carefully polished, e.g., some grammar errors, the layout of Figures and Tables.
7.    The submitted code cannot re-implement the results in this paper, and there is a large performance gap. So I recommend to reject this paper.

[1] Kurmi V K, Subramanian V K, Namboodiri V P. Domain Impression: A Source Data Free Domain Adaptation Method[C]//Proceedings of the IEEE/CVF Winter Conference on Applications of Computer Vision. 2021: 615-625.
[2] Tian J, Zhang J, Li W, et al. VDM-DA: Virtual Domain Modeling for Source Data-free Domain Adaptation[J]. arXiv preprint arXiv:2103.14357, 2021.
[3] Yeh H W, Yang B, Yuen P C, et al. SoFA: Source-data-free Feature Alignment for Unsupervised Domain Adaptation[C]//Proceedings of the IEEE/CVF Winter Conference on Applications of Computer Vision. 2021: 474-483.

**Time Spent Reviewing:**

35h

---

> ### Author Response · Authors · 2021-08-10
> **Response to Reviewer 4Suf**
>
> We really appreciate that the reviewer took the time to consider the submitted code, please find our response below.
>
> **1. Please give more explanations and discussions about the motivation of utilizing the memory bank to retrieve nearest neighbors.**
>
> We use the memory bank to have access to the feature location of all target data points during the training. Without the memory bank we would only have access to the data in the minibatch, and since we want to find the nearest neighbors considering all target  data we use a memory bank. The values in the memory bank are not based on the latest model, since data is only updated once per epoch, however, our results show that we can still obtain good results by using such a memory bank. Memory banks have been also applied in other methods [a,b]. We will improve the description of the memory bank.
>
> **2. Is the hyperparameter r used in Eq.5 and Eq.9 the same? Why not choose different hyperparameters for these two parts?**
>
> The hyperparameter $r$ is the same in Eq 5 and 8. Considering different hyperparameters could further improve results. For example, if we set the affinity of nRNN to 0.2 and the affinity of the expanded neighbors to 0.1, we can get a 0.3% improvement on VisDA. However, to not introduce extra hyperparameters, we keep the same $r$ for both equations.
>
> **3. Please provide comparative experiments with more source-free domain adaptation (SFDA) methods, such as [1] [2] [3].**
>
> [1] only reports result on Office-31, which is lower significantly than ours (83.5% versus 89.4%). [3] does not follow the mainstream experiment setting, they adopt AlexNet as backbone instead of ResNet which is used by almost all recent methods. [2] is an arxiv paper which is based on synthesizing virtual domain by Gaussian Mixture Model, and our method outperforms [2] on 2 of the 3 datasets, where results of [2] on Office-31, VisDA and PointDA-10 are 89.7%, 85.1% and 49.7% respectively, while our results are 89.4%, 85.9% and 52.6%. We will add discussion of these papers and their results into any final version of our paper.
>
> **4. It’s recommended to provide visualization of target features before adaptation in Fig.5 for better comparison.**
>
> The t-sne visualizations after adaptation are given in Figure 5 (right). For this experiment, the t-sne visualization before adaptation is given in Figure 1(a). We mention this in line 269.
>
> **5. Does the performance improve significantly at the expense of consuming large computational time cost? Please provide time complexity analysis of the proposed method.**
>
> We provide the runtime analysis below which shows that our method is faster than SHOT, please check our response (part [a]) to reviewer meMo for more details.
>
> | VisDA                         | **runtime (s/epoch with one TITAN Xp)** | Acc (%) |
> | ----------------------------- | --------------------------------------- | ------- |
> | SHOT                          | 618.82                                  | 82.9    |
> | **Ours**                      | 540.89                                  | 85.9    |
> | **Ours(20% for memory bank)** | 507.15                                  | 85.3    |
> | **Ours(10% for memory bank)** | 499.49                                  | 85.2    |
> | **Ours(5% for memory bank)**  | 492.28                                  | 85.1    |
>
>
> **6. The current manuscript should be carefully polished, e.g., some grammar errors, the layout of Figures and Tables.**
>
> Thanks for pointing out the typos, we will carefully revise any final version of the manuscript.
>
> **7. The submitted code cannot re-implement the results in this paper, and there is a large performance gap.**
>
> We really appreciate that you took the effort to consider the submitted code. In preparing the submitted code, we cleaned up our original code and accidentally omitted some functionality. Therefore, we provide a new anonymous **[link](https://anonymous.4open.science/r/anonymous_code-6B3C/readme.md)** that can reproduce the results in the paper. It contains the code and training log file along with the network weights (before and after adaptation). This code allows to reproduce results on VisDA and PointDA-10.
>
> *reference:*
>
> [a]Saito, Kuniaki, et al. "Universal Domain Adaptation through Self Supervision." NeurIPS 2020.
>
> [b]Liang, Jian, Dapeng Hu, and Jiashi Feng. "Domain Adaptation with Auxiliary Target Domain-Oriented Classifier." CVPR 2021.

---

> ### Comment · Reviewer_4Suf · 2021-09-01
> **Thanks for your response**
>
> Thanks for your response. This response cannot address my main concerns, and  I will keep my score. Thanks.

---

### Official Review · Reviewer_R8cb · 2021-07-18

**Rating:** 6
**Confidence:** 4

**Summary:**

This paper addresses the problem of Source Free Domain Adaptation for the classification task. SFDA is a challenging DA scenario where instead of source data only the source pretrained model is available for adaptation to unlabeled target.

This work focuses on closed-set domain adaptation task. The approach is motivated from an interesting observation that though targe samples do not align with the source classifier, these samples still form clusters with consistent class semantics. Following this, the authors aim to leverage this local neighborhood structure by encouraging label consistency among samples with high neighborhood affinity. The affinity resembles the degree of connectivity which is defined separately for general neighbors (nRNN), reciprocal neighbors (RNN), and extended neighbors (i.e., neighbor of neighbor).

The approach involves maintaining two memory banks, a) features and b) prediction scores for all the target samples. While neighbors are selected based on feature space similarity, the adaptation losses aim to maximize the affinity weighted prediction score similarity among the neighbors.


**Limitations And Societal Impact:**

No. The authors have not discussed any limitations of the approach. Certain limitations would be a) robustness to larger domain shifts, b) robustness to class-imbalanced target, etc. Further, the authors’ response of No to whether this approach could have potential negative societal impacts is a major oversight.

**Main Review:**

Strengths:

+The authors introduce a source-free domain adaptation method by using the intrinsic data structures. They propose a method for adaptation by encouraging label consistency among local target features by differentiating between nearest neighbors, reciprocal neighbors, and expanded neighbors, and further exploiting that information.

+The paper is well-written with clear justification and ablation analysis of the various learning components. The approach achieves significant improvement on VisDA-C (3%)  and marginal improvement (0.4%) over the prior-art [21] on the Office-Home dataset.


Weaknesses:

1) The minibatch update of $p_i$’s coupled with the self-regularization loss $L_{self}$ seems similar to an mommnetum update of $S=[p_1, p_2,..., p_{n_t}]$. While $p_i$ is computed based on the current model weights, $L_{self}$ does not allow drastic changes in $p_i$, thus acting as a regularizer. I am curious wether $L_{self}$ can be replaced by a simple momentum update of $S$.

2) Following the previous, any comments on what would be the behavior if such momentum update is applied while updating $F$. As I see, all the affinity-based losses and self-regularization is applied on $p$ while $z$ is used only for neighborhood selection (i.e. for assigning affinities). What if we apply similar regularization on $z$.

3) The proposed approach does not employ any pseudo-label-based loss as used in other prior arts. I am curious whether such a loss would further improve the adaptation performance. I observe that all the losses are defined on $p$, i.e., a softer form of pseudo-label.

4) I am curious about the importance of the diversity loss, $L_{div}$. I would like to see the performance where all other losses are active but without the diversity loss, $L_{div}$ (in the Left and middle panel of Table 5). This would give us some idea regarding robustness of the proposed approach against imbalanced target.

5) The authors claim that “source-free methods outperform methods that have access to source data during adaptation” (L215). They should have compared the same method with supervision from source data during adaptation. I would like them to show comparisons with other relevant non-SF prior-arts like A1. Also, can the performance of the proposed approach be improved further in the presence of source samples?
6) The authors have set the affinity value for the expanded neighbors to an arbitrary value of 0.1 (L254). And, the authors stated that expanded neighbors should have less affinity than the nearest neighbors (L165), yet they are using the same affinity, 0.1 for both expanded neighbors and nRNNs (L143, L254). Please clarify.

References:
[A1] Na et al, FixBi: Bridging Domain Spaces for Unsupervised Domain Adaptation [CVPR’21]


**** Post-rebuttal ****

I thank the authors for the detailed rebuttal. The rebuttal addresses most of my concerns regarding the limitation which were not discussed before. Below I restate some of these:
a) In presence of source data the performance of the method degrades.
b) Without diversity loss performance degrades significantly. This raises concerns about whether this method is at all suitable for open-set, partial, or universal DA settings.
c) The significance of setting different affinities for the expanded neighbors and nRNNs must be clarified thoroughly.

I hope that the authors would include a thorough discussion on the limitations of this approach in their revised draft. I'd like to raise the score to '6: Marginally above the acceptance threshold'.



**Time Spent Reviewing:**

10 hours

---

> ### Author Response · Authors · 2021-08-10
> **Response to Reviewer R8cb**
>
> Thanks for your comments, we will improve the paper to include all the discussions below.
>
> **1. Moment updating on prediction score bank may do similar job as the self-regularization**
>
> Indeed, in Eq (3) the prediction of the data point $i$ is not taken into account, we therefore introduce the self-regularization (which can be interpreted as a regularization). Using a momentum on the memory bank could result in a more stable update for prediction scores, but Eq. (3) would still ignore the prediction at $i$, and we do not think it could be used as a replacement of Eq. (6).
>
> **2. Also moment updating on feature bank $\mathcal{F}$ may help, and deploy similar loss on features.**
>
> Eq(3) aims to prompt similar class predictions for target data points that have high affinity scores (these affinity scores are based on the locations in the features space). We do not think that additionally putting such a loss on the features itself will provide much additional gain: since imposing similar feature locations to points with high affinity is probably already true, since these are computed based on the feature position. To test this, we have applied momentum to update $\mathcal{F}$. Results with moment updating (0.9 for old items) are almost the same as current results (0.1% improvement on VisDA). With respect to imposing additional losses on $z$: since we are using the fully connected layer classifier to give the prediction, we found additionally putting the related loss on feature $z$, which also aims to cluster target features, did not improve results.
>
> **3. The proposed approach does not employ any pseudo-label-based loss as used in other prior arts**
>
> Indeed, we do not apply any hard pseudo-labels (as for example SHOT). Combining the strength of papers that use pseudo-labels and ours could potentially further improve results. We consider this however outside the scope of the current work.
>
> **4. The importance of the diversity loss**
>
> The diversity loss is a widely used loss in unsupervised clustering and also domain adaptation [21, 37, 40] as mentioned in line 152, note the SRDC[21] and SHOT[40] also uses this loss. We provide the results without diversity loss below, and the SHOT also reports results without diversity loss in Table.6 of their paper, which is **63.3%** and much lower than our 76.3%. This loss is necessary since SFDA is totally unsupervised learning and easy to fall into a degeneration solution, where every sample is assigned to the same class. The author is right in pointing out that this loss might be less effective on imbalanced datasets. We will mention this in potential disadvantages of the method (we share this disadvantage with other methods that use this loss).
>
> | $L_{div}$ | $L_N$   | $L_E$   | $A$     | Avg  |
> | --------- | ------- | ------- | ------- | ---- |
> | $\surd$   | $\surd$ | $\surd$ | $\surd$ | 85.9 |
> |           | $\surd$ | $\surd$ | $\surd$ | 76.3 |
>
> **5. Comparing to paper A1. Can the performance of the proposed approach be improved further in the presence of source samples?**
>
> Different methods are designed for the different DA settings that consider having access to source data or not. It is therefore not straightforward to run a SFDA method with source data. However, our sentence L215 confirmed an observation already made in SHOT that showed that the methods designed for SFDA actually outperformed recent methods that do use source data. We will include the CVPR 2021 paper A1 (CVPR 2021 is held after the NeurIPS submission deadline) in the paper, and adapt the claim in sentence L215.
>
> Additionally, we have performed an experiment where our SFDA method has access to source data (for cross entropy loss) on VIsDA, and we observe in this case the target performance will degrade continually in the whole adaptation process, from 82.6% in the first epoch to 52.5% at the end. This result shows that our SFDA method is not made to exploit source data. Our SFDA method actually never aims (and is unable) to alleviate the domain shift, due to the absence of source data. So when providing source data, the performance of our method will degrade. The possible reason is that most SFDA methods like SHOT and ours are trying best to fully exploit the information of only the target domain. When applied with source data, the information from the source domain will impede the target adaptation of the SFDA method due to domain shift. As for normal DA methods, since most of them are trying to alleviate the domain shift, the source data are necessary.
>
> **6. The authors have set the affinity value for the expanded neighbors to an arbitrary value of 0.1 (L254). And, the authors stated that expanded neighbors should have less affinity than the nearest neighbors (L165), yet they are using the same affinity, 0.1 for both expanded neighbors and nRNNs (L143, L254). Please clarify.**
>
> The affinity values of expanded neighbors are the same as the nRNNs, since both types of these neighbors are expected to be less important than RNNs. If we set the affinity of nRNN to 0.2 and of the expanded neighbors to 0.1, we can get a 0.3% improvement on VisDA. To avoid introducing more hyperparameters and hyperparameter searches, we use the same affinity values.
>
>
> We also provide the result of SHOT for PointDA-10, which is significantly lower than ours.
>
> | PointDA-10 | Avg  |
> | ---------- | ---- |
> | SHOT       | 42.5 |
> | Ours       | 52.6 |
>
> And we will add the Limitations And Societal Impact part in the paper.

---

### Official Review · Reviewer_meMo · 2021-07-20

**Rating:** 7
**Confidence:** 4

**Summary:**

This paper introduces a neighborhood affinity based consistency approach for source-free domain adaptation. While some prior adaptation methods use nearest neighbors to obtain pseudo labels for the unlabeled target samples, this work proposes the use of reciprocal nearest neighbors (RNN). The observation is that RNNs show a higher degree of label consistency than nearest neighbors. Accordingly, the adaptation is formulated as improving the label consistency with the Reciprocal Nearest Neighbors of a given target instance. The method yields excellent performance over 4 datasets.

**Limitations And Societal Impact:**

I could not find a discussion on potential limitations and societal impact. I would suggest the authors add a discussion on these aspects. As an example, the nearest neighbor calculation and memoization operations are compute-heavy, which might put organizations with access to large computational resources at a higher advantage than others.

**Main Review:**

Strengths
========

a. The motivation for various nearest neighbor strategies is clear (L42-51). The proposed reciprocal nearest neighbor approach is intuitive and makes sense. The simplicity of the approach is a key highlight of this work.

b. Good work on the analysis. The figures are helpful in understanding the effectiveness of the approach. Figure 4 (Left) presents another interesting observation where per-shared fraction (measured without using target labels) converges with target accuracy (more details in the next section). The ablation study is detailed and supports the claims made in the paper. Moreover, the insights presented in this work would be highly useful for research in Source-Free Domain Adaptation.

c. The manuscript is mostly easy to follow (though some clarifications and typographical fixes are required). Overall, it is a good read.

d. The inclusion of code is appreciated.


Scope of improvement
=================

a. Scalability: The requirement of a memory bank and nearest neighbor computation seems like a large overhead (e.g. consider large scale datasets like ImageNet). While I understand that features $z_i$ and prediction scores $p_i$ are cheaper to memoize than the input data itself, it is imperative to investigate the effect of reducing the space and computational overhead.

* A runtime analysis (training time, FLOPS / epoch etc.) is required to understand the scalability of the proposed framework. Could the authors provide a runtime comparison against the prior state-of-the-art methods?

* Could the authors provide more details on how the nearest neighbors are computed? A typical nearest neighbor computation would entail $O(n_t * n_t)$  operations (quadratic) for each epoch (every target feature is compared with every other feature to obtain the list of nearest neighbors). In my opinion, this is a huge overhead considering that most adaptation algorithms require $O(n_t)$ operations (linear).

* Consider the following approach: Suppose the target training dataset is split into, say, 10 mutually exclusive and exhaustive folds. Each fold pertains to 10% of the data (call this, one mini-epoch) and is trained upon sequentially, i.e., we reduce the memory bank to 10% of the data, and ensure that samples in this mini-epoch pertain to the corresponding fold. After one epoch on the entire dataset is completed (i.e. after 10 mini-epochs), the data is randomly shuffled to create the next 10 folds. In this manner, the model is trained on the entire dataset, however, would require 10 times less overhead for memoization and NN-computation. Furthermore, this could reveal that perhaps only a fraction of the dataset in-memory is sufficient and the performance gain saturates with an increase in samples. If this works, then one of a major drawbacks of the proposed framework could be solved. Could the authors verify this and comment on the efficacy of this approach?

b. Writing:

* L147-149: This part is confusing. My understanding is that $\mathcal{S}\_{i}^{T}$ is a constant vector (not “scalar”, L147) containing the prediction scores of $x\_i^t$. The loss $\mathcal{L}\_{self}$ essentially computes a weighted sum (weights = $\mathcal{S}\_i$) of the class probabilities ($p\_i$). The expression is written like so (and not $p\_i^T \cdot p\_i$) because the loss $\mathcal{L}\_{self}$ is assumed to back-propagate only through the variable $p\_i$ and not through the weights ($\mathcal{S}\_i$). If this is the case, please consider paraphrasing this part to clarify the same.

* Some additional improvements are suggested in Minor Comments below.

c. Affinity measure: In the current formulation, affinity is a hyperparameter ($A = 1$ or $r<=1$). Thus, affinity is assigning discrete weights to the similarity between pairs of samples in Eq. 1. Did the authors consider using continuous affinity values to obtain a “degree of connectivity” (L55), for e.g. using feature distance $D_{dis} = || z_i - z_j ||$)?

d. The observation in Figure 4 (left) could potentially lead to a reliable convergence criterion. The usual practice in Unsupervised DA is to consider a labeled validation set for determining training convergence / selecting maximum epochs etc. A recent work [P1] presents an approach to determine training convergence without using target labels. In the proposed framework, the per-shared plot in Figure 4 (left) seems to correlate with the target accuracy. This could potentially be used as a convergence criterion. Could the authors provide more insights on this aspect?

[P1] Venkat et al., "Your Classifier can Secretly Suffice Multi-Source Domain Adaptation", NeurIPS 2020.


Minor Comments
=============

* A recent work by Wang et al., “Tent: Fully Test-Time Adaptation by Entropy Minimization”, ICLR 2021 also performs source-free adaptation using entropy minimization which results in a simple approach. It would be interesting to compare the runtime vs. performance trade-off between Tent and the proposed neighborhood approach.

* Typos: L273 (consistence -> consistency), L217 (date -> data), L145 (simple -> simply), L115 (weight -> weigh), L103, 232 (discusses -> discussed), Fig. 1 caption (pushed -> pushing), L34 (vary -> very).

* L90, L149, L270: Avoid 1-3 worded lines to save space (paraphrase the sentence).

* L45-51: Here, it would be a good idea to introduce / define RNN and nRNN so that the figure 1(c) is comprehensible. Also, in Fig. 1(c) left, consider showing nearest neighbors connectivity on the left and the RNN/nRNN plot on the right (i.e. swap the two diagrams). This would enhance readability.


I would be happy to increase the score if my concerns are addressed.

**Time Spent Reviewing:**

8

---

> ### Author Response · Authors · 2021-08-10
> **Response to Reviewer meMo**
>
> Thanks for all the advices, we will include the discussion below in any final version.
>
> **[a] - 1 and 2. Runtime analysis and more details on how the nearest neighbors are computed**
>
> In every mini-batch, we first do K-nearest neighbor retrieval for all target features in the mini-batch. Then for each of the K-nearest neighbors we further retrieve its M nearest neighbors. Note the competitive SFDA methods either synthesize labeled images which is hard to train and quite time-consuming (ModelAdaptation), or compute the pseudo label by using all target features every few iterations during adaptation (SHOT). We compare the runtime for one epoch of SHOT (including their proposed pseudo labeling) and our methods, specifically for SHOT the pseudo label is only computed once per epoch for all target samples, the results are below, where our method is faster than SHOT. Note we do not use any accelerated library for nearest neighbor retrieving (which would reduce complexity to $\mathcal{O}(n_t\ log n_t))$, which can be utilized to further speed up the training. Note that VisDA is the largest dataset (compared to Office-31 and Office-home).
>
> | VisDA                         | **runtime (s/epoch with one TITAN Xp)** | Acc (%) |
> | ----------------------------- | --------------------------------------- | ------- |
> | SHOT                          | 618.82                                  | 82.9    |
> | **Ours**                      | 540.89                                  | 85.9    |
> | **Ours(20% for memory bank)** | 507.15                                  | 85.3    |
> | **Ours(10% for memory bank)** | 499.49                                  | 85.2    |
> | **Ours(5% for memory bank)**  | 492.28                                  | 85.1    |
>
>
> **3. Trying to decrease the size of memory banks**
>
> Thanks for the advice. Here we use another simple way to decrease the storing memory. Instead of storing all feature vectors in the memory bank, we follow the same memory bank setting as in [2] which is for nearest neighbor retrieval. The method only stores a fixed number of target features, we update the memory bank at the end of each iteration by taking the n (batch size) embeddings from the current training iteration and concatenating them at the end of the memory bank, and discard the oldest n elements from the memory bank. We report the results with this type of memory bank of different buffer size in the table above. The results show that indeed this could be an efficient way to reduce computation on very large datasets.
>
> **[b] Writing related to self-regularization**
>
> Thanks for your comment on the unclear parts and we will improve this part in the final version. The self-regularization loss is exactly what you mention; the gradient is only coming from the class probabilities ($p_i$).
>
> **[c] Using continuous affinity values**
>
> Thanks for the advice. Without considering reciprocal neighbors, we here also assign affinity based on the similarity between the feature and its neighbors, for feature $z_i$​​​, inspired by instance discrimination [3] the affinity value of its $j$​​​-th $K$​​​-nearest neighbor $z_{ij}$​​​ is defined by: $A_{ij}=\frac{K\cdot exp(z_i^T z_{ij})}{\sum_{k=1}^K exp(z_i^T z_{ik})}$​​​​​​. Note that here the degree of connectivity is a continuous number. However, this approach gets 3.1% lower performance on VisDA than using reciprocal neighbors as in the paper. Note the core of our method is reciprocity (neighbors and neighbors of the neighbors), so simply considering similarity/distance between the feature and its neighbors ignores the reciprocity information, and it is nontrivial to combine reciprocity and continuous affinity. This could be a future direction of using neighbors for SFDA.
>
> **[d] The per-shared plot in Figure 4 (left) seems to correlate with the target accuracy. This could potentially be used as a convergence criterion. Could the authors provide more insights on this aspect?**
>
> Thanks for pointing out this finding. The evolution of the per-shared curve shows that the target features are forming clusters during the training. Further, it may imply that the SFDA can be regarded as an unsupervised deep clustering [1] problem. There are two mainstream directions of deep clustering, namely simultaneously or alternatively learning the feature representation and clustering assignment. SFDA is similar to the latter, where we first pretrain the model on the source domain (representation learning) and then do clustering. Compared to deep clustering, SFDA 1) has labeled source data for training and 2) category information (number and order which is dependent on the source classifier). We leave a deeper analysis of converge analysis based on per-shared features (motivated from deep clustering research) to future research.
>
> **other comments:** We will improve the paper carefully (typos, figures, the Limitations And Societal Impact part) in any revised version. And for Tent, it is designed more specifically for online setting where the target data is only used once, since many papers already try entropy minimizing [4,5] in DA (though they do not fix most parts). Tent is expected to be faster but get lower results in source free setting. We will include the discussion in the paper.
>
> *reference:*
>
> [1] Caron, Mathilde, et al. "Deep clustering for unsupervised learning of visual features." ECCV 2018.
>
> [2] Dwibedi, D., Aytar, Y., Tompson, J., Sermanet, P., & Zisserman, A. (2021). With a little help from my friends: Nearest-neighbor contrastive learning of visual representations. arXiv preprint arXiv:2104.14548.
>
> [3] Wu, Zhirong, et al. "Unsupervised feature learning via non-parametric instance discrimination." CVPR 2018.
>
> [4] Shu, Rui, et al. "A DIRT-T Approach to Unsupervised Domain Adaptation." ICLR 2018.
>
> [5] Liang, Jian, Dapeng Hu, and Jiashi Feng. "Do we really need to access the source data? source hypothesis transfer for unsupervised domain adaptation."ICML 2020.

---

### Author Response · Authors · 2021-08-05
**Code for reproducing results**

We want to thank Reviewers for reviewing our manuscript carefully.  Before entering into the questions in more detail, we would like to already respond to the question about the code (reproducibility). We really appreciate that Reviewer 4Suf took effort to consider the submitted code. In preparing the submitted code, we cleaned up our original code and accidentally omitted some functionality. Therefore, we provide a new anonymous [**link**](https://anonymous.4open.science/r/anonymous_code-6B3C/readme.md) that can reproduce the results in the paper. It contains the code and training log file along with the network weights (before and after adaptation). This code allows to reproduce results on VisDA and PointDA-10.

---

### Decision · Program_Chairs · 2021-09-28

**Decision:**

Accept (Poster)

**Comment:**

This work proposes a framework to perform Source-free Domain Adaptation by fully exploiting the local representation structure in the latent space. Specifically, it maintains two memory banks for logits and representations, respectively. Using the a sample in the mini-batch as query, it retrieve semantically similar neighborhoods from the bank, and constrains their logits to be similar. In addition, a self-regularization loss is applied to mitigate noisy neighbors, and a diversity loss is used to balance the sizes among different clusters. The reviewers found the paper to be technically sound and the approach interesting. There were some concerns about the technical novelty and the strength of the results, for which the paper can be argued from both a positive and negative perspective, since it builds on previous work (reciprocal NNs) and thus have some incremental flavor. Nevertheless, the positive reviewers highlighted the novelty of the method as a whole. After discussion it was concluded that the paper merits publication.

**Consistency Experiment:**

NeurIPS has a long history of experimentation. In 2014, NeurIPS ran an experiment in which 10% of submissions were reviewed by two independent committees to quantify the randomness in the review process. This year, we repeated a variant of this experiment to see how the quality of the review process has changed over time.  This paper was part of the experiment and was therefore assigned to two committees (consisting of reviewers, an Area Chair, and a Senior Area Chair) that reached independent decisions.  If both committees made the same recommendation, this recommendation was followed. If a single committee recommended acceptance, the paper was accepted (with the exception of a few cases in which the other committee identified what we considered a fatal flaw, e.g., an error in a key result).

Both committees reached the same decision: **Accept (Poster)**

The other committee assigned to the paper recommended **Accept (Poster)**.  You can find the other set of reviews, along with any follow up discussion with the authors here:
https://openreview.net/forum?id=yhjpeuWepoj